# Harnessing the Diversity of *Burkholderia* spp. Prophages for Therapeutic Potential

**DOI:** 10.3390/cells13050428

**Published:** 2024-02-29

**Authors:** Hayley R. Nordstrom, Marissa P. Griffith, Vatsala Rangachar Srinivasa, Nathan R. Wallace, Anna Li, Vaughn S. Cooper, Ryan K. Shields, Daria Van Tyne

**Affiliations:** 1Division of Infectious Diseases, University of Pittsburgh School of Medicine, Pittsburgh, PA 15213, USA; 2Department of Microbiology and Molecular Genetics, University of Pittsburgh School of Medicine, Pittsburgh, PA 15213, USA; 3Center for Evolutionary Biology and Medicine, University of Pittsburgh School of Medicine, Pittsburgh, PA 15213, USA

**Keywords:** *Burkholderia*, prophage, antibiotic resistance, phage therapy

## Abstract

*Burkholderia* spp. are often resistant to antibiotics, and infections with these organisms are difficult to treat. A potential alternative treatment for *Burkholderia* spp. infections is bacteriophage (phage) therapy; however, it can be difficult to locate phages that target these bacteria. Prophages incorporated into the bacterial genome have been identified within *Burkholderia* spp. and may represent a source of useful phages for therapy. Here, we investigate whether prophages within *Burkholderia* spp. clinical isolates can kill conspecific and heterospecific isolates. Thirty-two *Burkholderia* spp. isolates were induced for prophage release, and harvested phages were tested for lytic activity against the same 32 isolates. Temperate phages were passaged and their host ranges were determined, resulting in four unique phages of prophage origin that showed different ranges of lytic activity. We also analyzed the prophage content of 35 *Burkholderia* spp. clinical isolate genomes and identified several prophages present in the genomes of multiple isolates of the same species. Finally, we observed that *Burkholdera cenocepacia* isolates were more phage-susceptible than *Burkholderia multivorans* isolates. Overall, our findings suggest that prophages present within *Burkholderia* spp. genomes are a potentially useful starting point for the isolation and development of novel phages for use in phage therapy.

## 1. Introduction

*Burkholderia* is a genus of gram-negative bacteria that encompasses nearly 50 different species [1]. These organisms are abundant in the environment, are found readily in soil and water, and are associated with the rhizospheres of several species of plants [2,3]. *Burkholderia* spp. are also opportunistic human pathogens, particularly the members of the *Burkholderia cepacia* complex (Bcc), a group of at least 20 species which possess high levels of intrinsic resistance to multiple classes of antibiotics [4]. Two Bcc species make up the majority of *Burkholderia* spp. clinical isolates in the United States: *Burkholderia multivorans* and *Burkholderia cenocepacia* [5,6]. While infections caused by *Burkholderia* spp. are relatively rare in healthy people, these bacteria cause difficult-to-treat infections in patients with compromised immune systems or chronic conditions such as cystic fibrosis (CF) and chronic granulomatous disease [7,8]. Chronic infection with the Bcc, and in particular *B. cenocepacia*, in CF patients is associated with increased morbidity and mortality, decreased lung function, and shorter life expectancy [4,8,9,10]. Treatment of *Burkholderia* spp. infections is further complicated by intrinsic resistance to many classes of antibiotics [11,12]. This makes *Burkholderia* spp. some of the most challenging bacteria to eradicate and increases the risk of transmission to vulnerable patients. Current treatment protocols for managing *Burkholderia* spp. infections typically include long courses of multiple antibiotics resulting in high rates of treatment failure [13]. Concern over these clinically challenging pathogens has led to a need for alternative treatment strategies.

Bacteriophage (phage) therapy is the use of naturally occurring viruses that infect bacteria to treat infections. Compassionate use of phage therapy has risen dramatically in the last decade, with numerous successful cases reported [14,15,16]. While phage therapy for *Burkholderia* spp. infections would be advantageous, to date, only three compassionate use cases of *Burkholderia*-targeting phages have been reported, with mixed results regarding clinical efficacy [17,18,19]. Studies investigating the use of phages to treat *Burkholderia* spp. infection in vivo have shown efficacy both in *Galleria mellonella* [20,21] and murine lung infection models [22,23]. Phage therapy for *Burkholderia* spp. infections nonetheless is currently limited by the large genomic diversity of the *Burkholderia* genus as well as the relatively small number of *Burkholderia*-targeting phages that have been isolated to date [5]. One alternative source of *Burkholderia*-targeting phages are prophages found within the genomes of *Burkholderia* spp. isolates [24]. This is because prophages that have integrated into bacterial genomes are the result of successful prior phage infection and therefore could be useful for phage therapy [25,26]. Engineering of prophages to render them obligately lytic and deploy them for phage therapy is a strategy that has been used previously, though not in *Burkholderia* [14]. Prophages found within the genomes of *Burkholderia* spp. may therefore represent a source of phages for potential therapeutic use.

In this study, we investigated whether temperate phages from *Burkholderia* spp. clinical isolates could lyse other clinical isolates belonging to the same (conspecific) or different (heterospecific) species. We induced prophage release from *Burkholderia* spp. clinical isolates using mitomycin C and propagated the phages that lysed alternative *Burkholderia* spp. isolates. We isolated four different phages with varying activities in this manner. Additionally, we characterized the prophage content of 35 *Burkholderia* spp. clinical isolates and explored associations between bacterial species, prophage content, and phage susceptibility. Taken together, this study represents a first step toward addressing the limited availability of *Burkholderia*-targeting phages, presents an alternative strategy for phage discovery, and uncovers valuable insights regarding prophage carriage among *Burkholderia* spp. clinical isolates.

## 2. Materials and Methods

### 2.1. Bacterial Isolates and Induction of Prophage Release

All *Burkholderia* spp. isolates used in this study were collected from patients at the University of Pittsburgh Medical Center (UPMC) as part of their routine clinical care. Most isolates were collected as part of the Enhanced Detection System for Healthcare Associated Transmission [27], and others were collected from patients being evaluated for compassionate use phage therapy. Isolate collection was approved by the University of Pittsburgh Institutional Review Board under protocols PRO07060222 and STUDY19110005.

To induce prophage release, 10 µL of stationary phase liquid culture of each bacterial isolate was inoculated into 5 mL of Luria Broth (LB) containing 2.5 µg/mL mitomycin C. Cultures were grown overnight with shaking at 37 °C. The next day, bacterial cells were pelleted, and liquid lysates were filtered through a 0.22 µm syringe filter to remove bacteria. The remaining lysates were presumed to contain phage particles that were released during growth.

### 2.2. Isolation of Phages and Host Range Testing

Lytic bacteriophage activity was identified with a soft agar overlay assay [28]. Briefly, square petri plates were prepared containing LB bottom agar (LB media with 1.5% agar, 1 mM CaCl_2_, and 1 mM MgCl_2_). Bacterial isolates were inoculated into LB media and incubated overnight at 37 °C. The following day, bacterial soft agar lawns were created by mixing 50 µL overnight bacterial culture with 5 mL LB top agar (LB media with 0.5% agarose, 1 mM CaCl_2_, and 1 mM MgCl_2_) cooled to 55 °C, plated on top of bottom agar plates, and allowed to solidify. After top agar bacterial lawns had solidified, 5 µL of potential phage-containing lysates were spotted on top of the lawn. Plates were incubated upright at 37 °C overnight. The next day, plates were examined for evidence of lytic phage activity in the form of a spot or clearing of the bacterial lawn. Clearance zones were “picked” with a pipette tip, transferred into 100 µL of suspension media (SM) buffer (50 mM TrisCl pH 7.5, 100 mM NaCl, 8 mM MgSO_4_), and were incubated overnight at 37 °C. Phage-mediated killing was confirmed by plating 10-fold serial dilutions of phage lysate onto a bacterial lawn in top agar, as above. Each phage was passaged four times before the generation of high-titer stocks.

To prepare high-titer liquid lysates of each phage, individual plaques picked after four rounds of plaque purification were transferred to 100 µL of SM buffer and were then mixed with 100 µL of overnight culture of the propagating bacterial isolate. The mixture was incubated at room temperature for 15 min, then mixed with 10mL of LB top agar, and overlaid onto large (15 cm) bottom agar plates and allowed to set. Plates were incubated overnight at 37 °C. Plates showing semi-confluent lysis were flooded with 10 mL of SM buffer and were incubated at 37 °C for 2 h. The SM lysate was then collected and centrifuged at 4000× *g* for 10 min to pellet bacteria. Supernatants were filter sterilized through a 0.22 µm membrane syringe filter and were stored for future use.

Host range testing was performed using the soft agar overlay spot screening method [28]. Briefly, 5 µL of 10-fold serial dilutions of each phage lysate was spotted onto top agar lawns of each bacterial isolate and incubated at 37 °C overnight. The following day, each phage-bacteria pairing was assessed for clearance of the bacterial lawn. For pairings where lysis was noted, phage titer in plaque-forming units (PFU)/mL was recorded.

### 2.3. Whole Genome Sequencing and Analysis

Bacterial genomic DNA was extracted from 1 mL overnight cultures grown in LB media using a Qiagen DNeasy Blood and Tissue Kit (Qiagen, Germantown, MD, USA) following the manufacturer’s protocol. Phage genomic DNA was extracted from aliquots of high-titer liquid lysate using the same kit or via phenol chloroform extraction followed by ethanol precipitation. Briefly, 500 µL phenol/chloroform/isoamyl alcohol (25:24:1) was added to 500 µL of each lysate, and samples were vortexed and then centrifuged at 16,000× *g* for 1 min. The upper aqueous phase was transferred to a new tube and 500 µL of chloroform was added. Samples were vortexed and centrifuged again at 16,000× *g* for 1 min, and the upper aqueous phase was again transferred to a new tube. Then, 1 µL glycogen, 0.1 × volume 3 M sodium acetate, and 2.5 × volume 100% ethanol were added, and samples were incubated overnight at −20 °C. The next day, samples were centrifuged at 16,000× *g* for 30 min at 4 °C, and then the supernatant was removed and the DNA pellet was washed with 150 µL 70% ethanol. DNA pellets were resuspended in 100 µL nuclease-free water. All DNA were quantified with a Qubit fluorimeter (Thermo Fisher Scientific, Waltham, MA, USA). Bacterial and phage genomes were sequenced on the Illumina platform at the Microbial Genome Sequencing (MiGS) Center (Pittsburgh, PA, USA). Illumina library construction and sequencing were conducted using an Illumina Nextera DNA Sample Prep Kit with 150 bp paired-end reads, and libraries were sequenced on the NextSeq 550 sequencing platform (Illumina, San Diego, CA, USA).

Genomes were assembled with SPAdes v3.11 to generate contigs with a 200 bp minimum length cut-off [29]. Phage contigs were extracted from each assembly and were separated from contaminating host bacterial sequences by examining the differential read coverage of each contig, and with BLASTN. Assembled genomes were annotated with RAST [30]. A core genome SNP alignment was generated with snippy v4.6 (https://github.com/tseemann/snippy, accessed on 16 November 2021), then the core alignment was used to construct a phylogenetic tree with RAxML v8.2.12 with the GTRCAT substitution model and 1000 iterations [31]. Bacterial species were assigned by average nucleotide identity comparisons with previously sequenced *Burkholderia* species using fastANI v1.33 [32]. Prophages were identified in each bacterial genome using PHASTER (https://phaster.ca/) [33]. Prophages of any length that were predicted to be intact, questionable, or incomplete by PHASTER were included. Prophage sequences were compared to one another with BLASTN v2.12 [34], and clusters of similar prophage sequences were identified as those sharing >90% sequence coverage and >90% sequence identity. A cluster analysis of all prophage sequences was performed and visualized using Cytoscape v3.7.2 [35]. Predicted family and genus of each bacteriophage was determined by the closest BLAST match in the NCBI nonredundant (nr) database.

### 2.4. EM Imaging

Between 1 and 5 µL of bacteriophage BCC02 suspension was added to a copper grid and negatively stained with 1% uranyl acetate. Phage suspension was imaged by transmission electron microscopy on a JEW 1400 Flash Transmission Electron Microscope. Imaging was performed by the University of Pittsburgh Center for Biologic Imaging (Pittsburgh, PA, USA).

### 2.5. Statistical Analysis

Statistical analyses were performed using GraphPad Prism v8.0.0 (GraphPad Software, San Diego, CA, USA). Linear regression and two-tailed *t*-tests were performed to assess significance of associations between prophage abundance and phage susceptibility. A Fisher’s Exact Test was used to assess the association between Bmulti_pp1 and phage resistance in *B. multivorans* isolates.

## 3. Results

### 3.1. Prophage Induction and Isolation

To test whether prophages found within the genomes of *Burkholderia* spp. clinical isolates could target other *Burkholderia* spp. isolates, we first created a library of potential phage-containing lysates. To do this, we inoculated 32 *Burkholderia* spp. clinical isolates collected from 28 unique patients into liquid culture in the presence of the mutagen mitomycin C, which prompts prophage excision (Figure 1) [36,37]. We then performed an all-by-all screen of the 32 phage-containing “source” lysates against the same 32 *Burkholderia* spp. “target” isolates, using a spot-plaque screening method to perform 1024 pairwise tests (Figure 1). For 11 of these tests, a phage-containing source lysate inhibited the growth of the target bacterial lawn (1.1% hit rate). Each positive pairing was retested to confirm that the observed inhibitory activity was due to phage activity by serially diluting the source lysate and looking for individual plaques. The 10 source–target pairings with confirmed phage activity were then subjected to four rounds of picking and passaging of single plaques to isolate individual phages. Three of these pairings did not maintain lytic activity through picking and passaging, meaning that during one of the passages, no plaques were visible. The seven remaining source–target pairs yielded viable phages, which were amplified into high-titer lysates and designated as BCC02 through BCC08 (Appendix A).

### 3.2. Whole Genome Sequencing of Burkholderia spp. Clinical Isolates

To explore the genetic diversity of the *Burkholderia* spp. clinical isolates studied here, we sequenced 35 bacterial genomes, including the 32 isolates used for prophage induction and testing, plus three additional clinical isolates. Isolates were sequenced on the Illumina platform and a core genome phylogeny was constructed (Figure 2). The species of each isolate was determined using fastANI [32] to compare the average nucleotide identity between each genome and available genomes of type strains of *Burkholderia* spp. The most frequently sampled species were *B. multivorans* (16 isolates collected from 13 patients) and *B. cenocepacia* (14 isolates collected from 13 patients). One isolate each belonged to *Burkholderia gladioli*, *Burkholderia pseudomultivorans*, *Burkholderia vietnamiensis*, *Burkholderia seminalis*, and *Burkholderia cepacia* (Figure 2). All isolates except DVT1600 (*Burkholderia gladioli*) fell into the *B. cepacia* complex (Figure 2).

### 3.3. Phage Host-Range Screening

To determine the infectivity profile of each isolated temperate phage as well as the phage susceptibility of each *Burkholderia* spp. clinical isolate, we performed an all-by-all phage activity screen. We tested all seven prophage-derived phages as well as two additional *Burkholderia*-targeting bacteriophages of environmental origin, DSMZ107315 and Bch7 [19], against all 35 *Burkholderia* spp. clinical isolates (Figure 2). Eight isolates, all of which were *B. multivorans*, were resistant to all phages tested. However, the other 27 isolates (77.1% of all isolates tested) were susceptible to at least one phage. We observed variability in the host range of each phage, with phages able to lyse between 2 (BCC06) and 12 (Bch7) of the 35 isolates. All phages were able to infect and lyse multiple isolates, and in most cases, phages were able to lyse bacteria belonging to different species (Figure 2). We observed similar host ranges for phages BCC02, BCC03, and BCC04, as well as for phages BCC05 and BCC06. This finding, along with the fact that these groups contained phages derived from the same donor bacterial isolates (Appendix A), suggested that these might represent duplicate isolations of the same phage. Whole genome sequencing confirmed this, as described further below. 

### 3.4. Whole Genome Sequencing of Phages

To explore the genetic diversity of the phages tested in this study, phage lysates were subjected to genomic DNA extraction and sequencing on the Illumina platform. High-coverage phage-derived contigs were analyzed and were compared to each other, to bacterial prophage genomes, and to publicly available genomes (Table 1). Phage genomes ranged in size from 23 to 68 kb and varied in GC content from 54.7 to 67.1% (Table 1). Phages BCC02, BCC03, and BCC04, which were all isolated from the DVT1180 source isolate, were found to be genetically identical to one another. Phage BCC02 therefore was chosen as the representative phage for follow-up work. Likewise, phages BCC05 and BCC06, which were both isolated from the DVT1166 source isolate, were found to be identical except for one single nucleotide polymorphism within a gene predicted to encode a phage tail fiber protein, suggesting divergent evolution during the course of phage isolation. However, as we observed similar host ranges between BCC05 and BCC06, for the purposes of our analysis, we also considered these two phages to be identical and chose BCC05 as the representative phage. The family and genus of each phage were predicted based on sequence comparisons with previously described phages deposited in the NCBI using nucleotide BLAST (Table 1). Transmission electron microscopy (TEM) imaging of phage BCC02 confirmed that it had a contractile tail and icosahedral head (Figure 3). Overall, genomic analyses revealed that our approach yielded four novel temperate bacteriophages targeting *Burkholderia* spp. (BCC02, BCC05, BCC07, and BCC08) that were distinct from each other and from other known phages.

### 3.5. Analysis of Prophage Carriage in Burkholderia spp. Clinical Isolates

To understand the diversity of prophages present in the *Burkholderia* spp. clinical isolates we collected, we extracted prophage sequences from the genomes of the 35 isolates used for screening. We searched each isolate genome for sequences of likely prophage origin using PHASTER [33]. A total of 114 prophage sequences were identified (range = 1–7, median = 3 prophages per genome) (Appendix A) (Figure 4a). We tested whether prophage abundance was associated with phage susceptibility and found that these two features were only loosely correlated with one another (linear regression *p* = 0.0675) (Appendix A). However, we did find that *B. multivorans* isolates contained more prophages compared with *B. cenocepacia* isolates (3.8 vs. 2.4 average prophages, *p* = 0.0266) (Figure 4b). Additionally, *B. multivorans* isolates were susceptible to fewer phages compared with *B. cenocepacia* isolates (*p* = 0.0005) (Figure 4c), in agreement with other studies, demonstrating an inverse correlation between prophage abundance and phage susceptibility in different bacterial species [38,39,40]. 

To determine the extent of relatedness among prophages encoded in *Burkholderia* spp. clinical isolate genomes, we compared all extracted prophage sequences to one another using nucleotide BLAST [34]. Prophages that shared >90% sequence coverage and >90% sequence identity were grouped into clusters, and clusters were visualized using Cytoscape [35] (Figure 5) (Appendix A). Twenty clusters of similar prophages were identified, which ranged in size from 2 to 4 with the exception of one large cluster containing 11 prophages. Overall, prophage clusters tended to group with related host strains. For example, our screening panel contained two sets of isolates that were gathered at different time points from the same patient; DVT1140, DVT1171, DVT1172, and DVT1181 (*B. multivorans*) were collected from one patient, while DVT599 and DVT1154 (*B. cenocepacia*) were collected from another. The prophages present in these isolates clustered together, as expected (Figure 5) (Appendix A). Isolated phages BCC02/BCC03/BCC04, BCC05/BCC06, BCC07, and BCC08 did not fall into any of these clusters. Finally, one cluster, which we named Bmulti_pp1, contained 11 prophage sequences from 10 *B. multivorans* isolates. Presence of this prophage was significantly associated with phage resistance among the *B. multivorans* isolates we tested (*p* = 0.035) (Appendix A). This finding suggests that Bmulti_pp1 may provide protection against phage-mediated lysis in *B. multivorans.*

## 4. Discussion

The establishment and maintenance of lysogeny in bacterial hosts by temperate phages is a widespread phenomenon that involves a complex interplay of elements both on cellular and ecological levels. These interactions are influenced by many factors, including genetics, cellular development, community dynamics, and environmental conditions [41]. In this study, we show that, in some cases, the relationship between bacteria and their prophages can be exploited to isolate bacteriophages with potential therapeutic use and that prophage induction with mitomycin C is an effective method to accomplish this in *Burkholderia* spp.

Polylysogeny in *Burkholderia* appears to be common [24], and prophages present in *Burkholderia* genomes likely represent a rich hunting ground for clinically useful bacteriophages [42]. Triggering the bacterial DNA damage response using a mutagen like mitomycin C is a simple way to activate the lysogenic-lytic switch for some prophages [43], and the prophages we induced with this method showed lytic activity against both conspecific and heterospecific bacterial isolates. However, further characterization and rational engineering of temperate phages usually is considered to be necessary before they can be used clinically. Engineering of *Burkholderia* prophages to render them obligately lytic, either through site-directed mutagenesis or experimental evolution, has been previously described [44,45]. Similar approaches could be taken with the prophage-derived phages described here to convert them from temperate to lytic phages. Such exploration of the use of temperate bacteriophages in phage therapy has the potential to expand the pool of useful tools against the escalating threat of multidrug-resistant bacteria when naturally occurring obligately lytic phages are rare. 

In contrast to prior studies [46,47], here we performed a systematic screening of lytic phage activity among induced prophages using more than 30 genetically diverse *Burkholderia* spp. clinical isolates. The four temperate phages we studied showed combined activity against 60% (21/35) of the clinical isolates that were tested. The isolated prophage BCC02, while a temperate phage, was able to lyse 31% (11/35) of the clinical isolates used in our study. This phage is genetically similar to the previously described *Burkholderia* phage KS5 isolated by Seed et al., which also demonstrated broad lytic activity [48]. The four novel temperate phages isolated and described in this study have potential for further development toward therapeutic application. With the exception of BCC05 and BCC06, the host ranges of phages of prophage origin were comparable to those of environmental origin, in agreement with the previous literature [48,49]. Additionally, identification of phages with varying host ranges and infection dynamics allows for the development of multi-phage cocktails in order to reduce the probability of phage resistance [46]. 

Our findings demonstrate that clinically significant *Burkholderia* isolates host a variety of prophage elements, in agreement with previous studies, showing that lysogeny is relatively common in this genus [3,49,50]. While our study was likely underpowered to identify a strong correlation between prophage abundance and phage susceptibility, we found more prophages and greater overall phage resistance in *B. multivorans* isolates compared to *B. cenocepacia* isolates. A prior study similarly noted the relative phage resistance of *B. multivorans* isolates [49]. We also identified one prophage that was associated with resistance to phage lysis, which may indicate some evolutionary or ecological relevance, possibly enabling superinfection immunity. The occurrence of this prophage could indicate ancestral integration maintained through a fitness advantage or direct phage transmission between clinical strains in close contact with one another. Exploring determinants of prophage infectivity and the effects of the Bmulti_pp1 prophage on infection dynamics of other phages require further study.

This study had several limitations. The small volume spot-screening method used to isolate induced phages may have missed phages that were present in lysates at low concentration. We did not investigate the efficiency of temperate phage induction or how well it correlated with the recovery of an active temperate phage. Additionally, slow-growing clinical isolates may have hindered the detection of lytic activity, since some temperate phages are known to induce lysis only when the host density is high [41]. All genomes used for our analysis were draft genomes, and many were not able to be completely assembled within the scope of this project. This could potentially skew our analysis of the number of prophages in each isolate, as some prophages may have spanned multiple contigs. Finally, all work in this study was performed in vitro, thus we are unable to conclude that these newly characterized phages would be useful for clinical therapy without further testing in a relevant in vivo model of infection.

## 5. Conclusions

Overall, the data generated in this study add to the literature characterizing *Burkholderia*-targeting bacteriophages, as well as prophage abundance and diversity in clinically relevant *Burkholderia* species. We isolated four novel phages with lytic activity against a variety of *Burkholderia* isolates, identified temperate phages with a broad host range, and detected an association between one particular prophage and phage resistance in *B. multivorans*. The isolated phages of prophage origin may also have clinical utility and are a potentially useful starting point for the development of novel phages for use in phage therapy.

## Figures and Tables

**Figure 1 cells-13-00428-f001:**
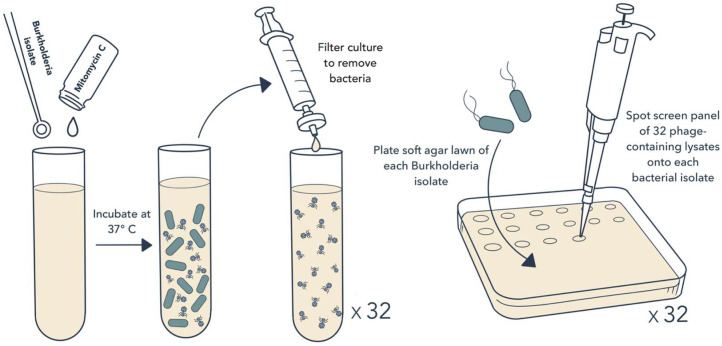
Approach for prophage induction and testing. To induce prophage release, stationary phase liquid cultures of 32 *Burkholderia* spp. clinical isolates were individually inoculated into LB media containing mitomycin C. Cultures were grown overnight at 37 °C. The next day, bacterial cells were pelleted and liquid lysates were filtered through a 0.22 μm filter. Filtered lysates were then spotted onto soft agar lawns containing each of the same 32 *Burkholderia* spp. clinical isolates. Plates were examined for growth inhibition, then inhibitory lysates were retested to confirm phage activity.

**Figure 2 cells-13-00428-f002:**
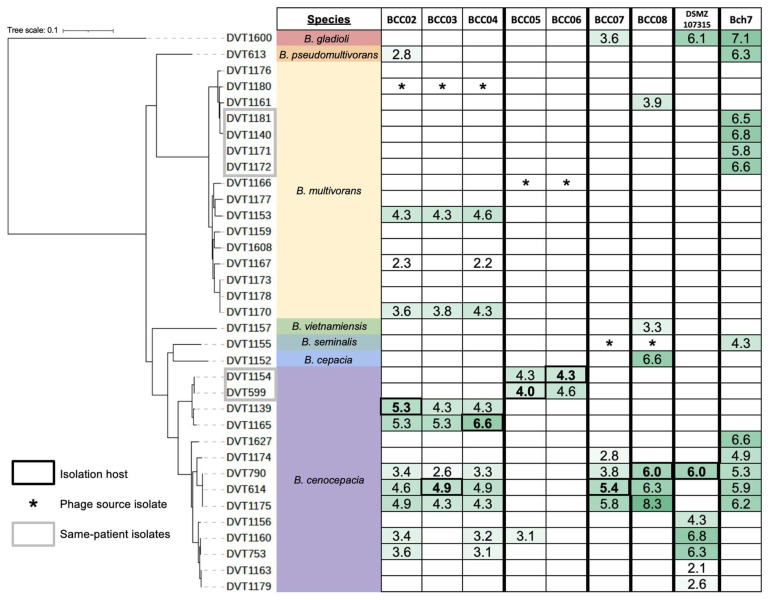
Phage activity against genetically diverse *Burkholderia* spp. clinical isolates. Bacterial isolates are ordered according to their core genome phylogeny and are grouped by species. Grey boxes indicate isolates collected from the same patient. Infectivity is shown as the log_10_ titer (PFU/mL) of each phage against each isolate. Bolded values indicate the *Burkholderia* spp. target isolate that each phage was isolated and propagated on. Asterisks mark the source *Burkholderia* spp. isolate for each phage. Green shading corresponds to phage activity titer, with darker shading indicating a higher titer. Empty cells indicate no phage activity.

**Figure 3 cells-13-00428-f003:**
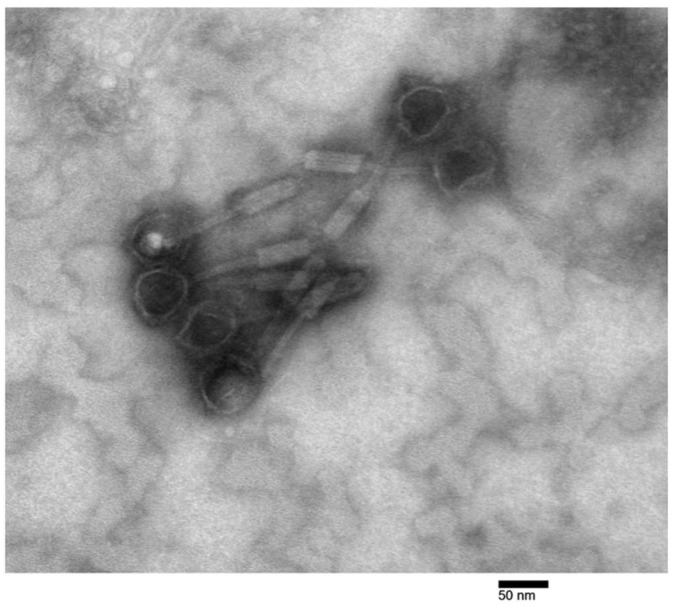
Electron micrograph of temperate phage BCC02. Transmission electron micrograph showing *Burkholderia* phage BCC02. Image was taken at 200,000× magnification. Icosahedral head, tail, and contractile tail sheath are visible for 6 virions.

**Figure 4 cells-13-00428-f004:**
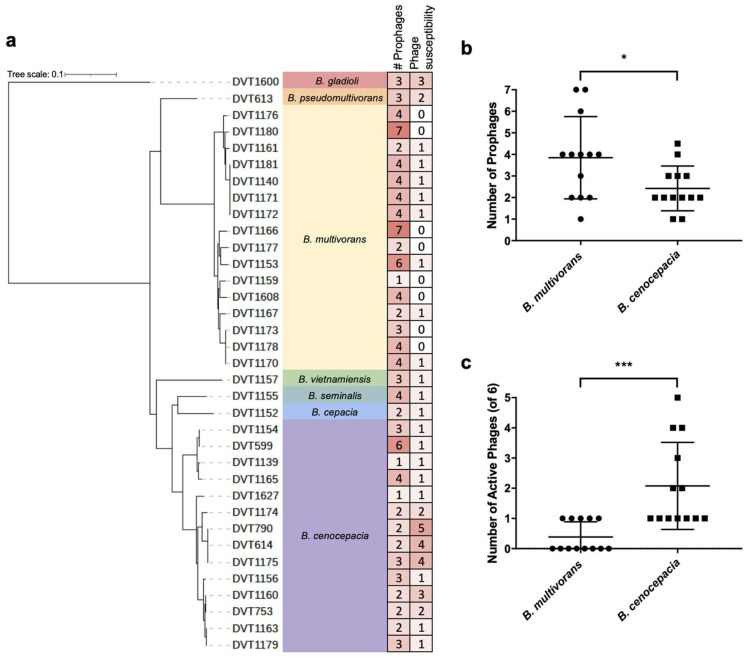
Prophage abundance and phage susceptibility of diverse *Burkholderia* spp. clinical isolates. (**a**) Core genome phylogeny of 35 *Burkholderia* spp. clinical isolates showing the number of prophages and phage susceptibility of each isolate. For phage susceptibility, BCC02/BCC03/BCC04 and BCC05/BCC06 were each combined into a single count. Darker red shading corresponds with higher values. (**b**) Prophage abundance among *B. multivorans* and *B. cenocepacia* genomes. (**c**) Phage susceptibility among *B. multivorans* and *B. cenocepacia* isolates. *p*-values are from unpaired two-tailed *t*-tests. * *p* < 0.05; *** *p* < 0.001.

**Figure 5 cells-13-00428-f005:**
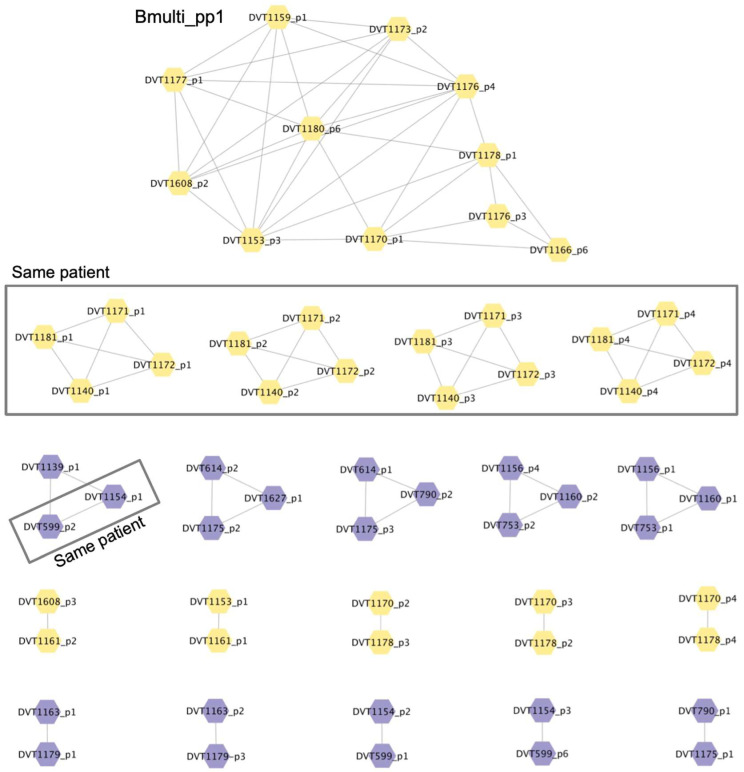
Clusters of prophages encoded by *Burkholderia* spp. clinical isolate genomes. Bacterial isolate names and prophage numbers are listed inside the nodes of each cluster, and lines connect prophages that share >90% sequence coverage and >90% sequence identity. Yellow nodes indicate prophages in *B. multivorans* isolate genomes and purple nodes indicate prophages in *B. cenocepacia* isolate genomes. Isolates from the same patient (two separate patients) are boxed.

**Table 1 cells-13-00428-t001:** Genome summary of *Burkholderia*-targeting phages.

Phage ID	Source Isolate	Source Species	Length (bp)	GC %	Predicted Genus ^1^	NCBI Similar Phage ^2^
BCC02/03/04	DVT1180	*B. multivorans*	34,126	64.2	*Peduovirinae;* *Kisquinquevirus*	*Burkholderia* Phage KS5 (GU911303.1)
BCC05/06	DVT1166	*B. multivorans*	30,957	62.8	*Peduoviridae;* *Duodecimduovirus*	*Burkholderia* Phage phiE12-2 (NC_009236.1)
BCC07	DVT1155	*B. seminalis*	38,216	66.7	*Peduoviridae;* *Aptresvirus*	*Burkholderia* Phage Mana (NC_055863.1)
BCC08	DVT1155	*B. seminalis*	28,709	63.7	*Peduoviridae;* *Kayeltresvirus*	*Burkholderia* Phage KL3 (GU911304.1)
Bch7	Env ^3^	-	68,166	54.7	*Bcepfunavirus*	*Burkholderia* Phage Maja (MT708549.1)
DSMZ 107315	Env ^3^	-	22,967	61.5	*Peduovirinae;* *Kisquattuordecimvirus*	*Burkholderia* Phage FLC5 (NC_055722.1)

^1^ Predictions were made based on nucleotide BLAST comparison with available phage genomes in NCBI. ^2^ Most similar phage identified in NCBI through nucleotide BLAST comparison of sequence coverage and identity. ^3^ Env = environmental.

## Data Availability

Whole genome sequencing of bacterial isolates and phages have been submitted to the NCBI under BioProject PRJNA1067351. Other data used in the manuscript are available in Appendix A. Bacterial isolates and phages are available from the corresponding author under the terms of a Materials Transfer Agreement.

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
