# Peer review of "Harnessing the Diversity of Burkholderia spp. Prophages for Therapeutic Potential"

_cells, 2024, doi:10.3390/cells13050428_

Round 1

Reviewer 1 Report

Comments and Suggestions for Authors

In this research article, Nordstrom et al. isolate a number of prophages from 32 different clinical isolates of Burkholderia spp.  Ultimately the authors have isolated and sequenced the genomes of 4 unique prophages.  These prophages are capable of infecting multiple Burkholderia spp however the underlying mechanism for specificity of infectivity remains unresolved.  The study is well-designed and executed, the description of the results and discussion are well constructed and appropriate.  The significance and impact are low.

Author Response

We thank the reviewer for their overall positive feedback on our work. We agree that investigating the mechanisms for the specificity of infectivity of the prophages we identified would be a useful next step. This will be a focus of our future work in this area, and we now note in the Discussion (lines 366-368).

Reviewer 2 Report

Comments and Suggestions for Authors

Nordstrom et al. describe the isolation and characterization of temperate bacteriophages targeting Burkholderia. The experimental results are solid and the manuscript is nicely written. This reviewer just has a few comments concerning the bacteriophage terminology used.

Specific comments:

Abstract, line 19: “Lytic phages were passaged …”. The bacteriophage nomenclature can be a bit confusing. Both lytic bacteriophages and temperate bacteriophages do have lytic activity. But the bacteriophage particles, generated by inducing a temperate bacteriophage (prophage) are not lytic bacteriophages. Thus, line 19 should read “Temperate phages were passaged …”

Introduction, lines 67-68: “… we investigated whether prophages from Burkholderia spp. clinical isolates could lyse other clinical isolates …”. Another small problem with the bacteriophage nomenclature. A temperate bacteriophage is only called a prophage when its genome is integrated into the chromosome of the bacterial host. Thus, a prophage does not have the ability to infect and lyse a bacterial isolate. Correct would be “… we investigated whether temperate phages from Burkholderia spp. clinical isolates could lyse other clinical isolates …”

Material and Methods, lines 93-94: “The remaining lysates were presumed to contain prophages that were released during growth.” As above, a phage lysate cannot contain a prophage. Correct would be “The remaining lysates were presumed to contain phage particles that were released during growth.”

Material and Methods, lines 105-106: “… evidence of lytic phage activity in the form of plaques or clearing of the bacterial lawn”. The authors describe a spot test analysis to detect bacteriophage activity. Such a test does not result in a plaque or plaques but rather in a spot. Thus better “… evidence of lytic phage activity in the form of a spot or clearing of the bacterial lawn”.

Material and Methods, lines 106-107: “Individual clear plaques were “picked” with …” As above, rather spots than plaques and were the “plaques” really clear? Temperate bacteriophages usually make rather turbid plaques as opposed to lytic bacteriophages, but that could be in the eye of the beholder. For the purpose of this manuscript, it would not make a difference either way. Suggestion: “Clearance zones were “picked” with …

Material and Methods, line 111: “…dilutions of prophage lysate to observe...” Same argument as above, just delete “pro”.

Material and Methods, line 132: “…was assessed for visibility of plaques.” Same argument as above, a spot test results in a spot rather than a plaque.

Results, line 189: “… prophage-containing lysates.” Same argument as above, just delete “pro”. The same goes for lines 192 and 194.

Results, lines 201-202: “…passages no lytic plaques were visible perhaps due to instability between lysis and lysogeny.” There are no lytic plaques, just plaques, and there is no instability between lysis and lysogeny so the explanation is rather nonsensical. But there is no need to find an explanation for the loss of phage activity at this stage of the manuscript. Just write: “… passages no plaques were visible.”

Legend to figure 1, line 206: Typo, should read “cultures”

Results, line 225: The term “prophage-derived phage” is not used, even if it is descriptive. The precise term would be “temperate phage” (i.e. a bacteriophage that has the ability to form an inducible prophage). In the context of this text also “induced phage” could be used.

Results, lines 265-267: The sentence “… yielded four novel Burkholderia-targeting bacteriophages (BCC02, BCC05, BCC07, and BCC08) of prophage origin that were distinct from each other and from other known phages.” is a bit clumsy, especially the prophage origin part. Better would be: “… yielded four novel temperate bacteriophages targeting Burkholderia (BCC02, BCC05, BCC07, and BCC08), that were distinct from each other and from other known phages.

Legend to figure 3, line 275: “Electron micrograph of prophage-derived phage BCC02.” In this context definitely use “temperate phage BCC02.“

Discussion, line 351: “… prophage-derived phages we studied” Better use “temperate phages” in this context as well.

Discussion, line 390: “… prophage-derived phages with broad…” Better use “temperate phages” here as well.

Author Response

We thank the reviewer for their careful reading of our manuscript and appreciate their expert input. We have made all the changes that they suggested, which are indicated with yellow highlighting in the revised manuscript.

Reviewer 3 Report

Comments and Suggestions for Authors

Authors in this manuscript describe in detail the process to isolate prophages with bacteriophage activity cross-specific in Burkholderia spp. The work is well-written, the data is novel, and the quality of the presentation is correct. They are aware of the limitations of the study, while they collected a significant amount of data. However, I suggest minor changes and some putative extra assays, as described below.

MINOR

- Line 54: I would recommend adding a brief summary explaining if the treatments were or were not successful or in which conditions.

- Line 55: authors mention Burkholderia spp. while the genus is very large, I think they mean Bcc, to differentiate from other complexes like pseudomallei-thailandesis

- Line 93-94: there is no option to check that through PCR, qPCR, or alike? Since authors finally sequenced all the strains, this induction could be checked by qPCR to see the efficiency of the release in order to standardize the process.

- Section 2.2: I find the section very long and detailed, but it may be reduced since many steps do not need the full description (for example: lines 101-103, line 107, lines 113-114, lines 122-123)

- Line 139-141: why did the authors use 2 different methods? Please if there is any reason, specify.

- Line 171: non-redundant?

- Line 174: One to 5, sentences cannot start with a digit

- Section 2.4 is very brief compared with the other sections. Please explain why that phage was the chosen one.

- Figures in general need quality improvement since they look pixelated, especially Figure 1.

- Line 217: remove (ANI)

- Line 223: B. cepacia

- Line 237-238: That was proved by sequencing, please indicate here )i.e., "as it was proved by sequencing, as further discussed")

- Line 253: BCC02, BCC03, and BCC04,

- Line 257: remove (SNP)

- Line 313: BCC05/BCC06, BCC07, and BCC08

- Have authors tried a cocktail mix of the lytic phages to overcome the endogenous resistance of some of the species? That would be a beneficial screening that could easily translate into clinics due to the strain-specific nature of some phage activity. If it was not performed I recommend an assay with the 7 active phages together for screening, and if it was please describe it.

Author Response

We thank the reviewer for their critical feedback. Here is a point-by-point response to their comments:

MINOR

- Line 54: I would recommend adding a brief summary explaining if the treatments were or were not successful or in which conditions.

-> We now note that clinical responses to Burkholderia phage therapy were variable.

- Line 55: authors mention Burkholderia spp. while the genus is very large, I think they mean Bcc, to differentiate from other complexes like pseudomallei-thailandesis

-> We decided not to use "Bcc" in favor of the broader "Burkholderia spp." because our study includes one B. gladioli isolate, which is not technically a member of the Bcc.

- Line 93-94: there is no option to check that through PCR, qPCR, or alike? Since authors finally sequenced all the strains, this induction could be checked by qPCR to see the efficiency of the release in order to standardize the process.

-> Thank you for this comment. We did not save the original temperate phage-containing lysates, and given that the all-by-all activity screen involved 32 different isolates, it would take quite a bit of time to go back and perform this work. We have nonetheless added this as an additional limitation in the Discussion section (Lines 371-372).

- Section 2.2: I find the section very long and detailed, but it may be reduced since many steps do not need the full description (for example: lines 101-103, line 107, lines 113-114, lines 122-123)

-> Thank you for this comment. We have streamlined this section, however the details that remain are meant to assist other researchers who wish to follow our methods. 

- Line 139-141: why did the authors use 2 different methods? Please if there is any reason, specify.

-> We first attempted to extract genomic DNA from phage lysates using a Qiagen kit. For some isolates this method yielded DNA samples with unacceptably low concentrations, in which case we reverted to extraction with phenol chloroform instead. 

- Line 171: non-redundant?

-> Correct, and this is now spelled out.

- Line 174: One to 5, sentences cannot start with a digit

-> This has been corrected.

- Section 2.4 is very brief compared with the other sections. Please explain why that phage was the chosen one.

-> We selected phage BCC02 for EM imaging because it had the broadest activity of all the temperate phages we studied.

- Figures in general need quality improvement since they look pixelated, especially Figure 1.

-> We defer to the editor and are happy to provide higher resolution figures if necessary.

- Line 217: remove (ANI)

-> Removed.

- Line 223: B. cepacia

-> Revised as suggested.

- Line 237-238: That was proved by sequencing, please indicate here )i.e., "as it was proved by sequencing, as further discussed")

-> Revised as suggested.

- Line 253: BCC02, BCC03, and BCC04,

-> Revised as suggested.

- Line 257: remove (SNP)

-> Revised as suggested.

- Line 313: BCC05/BCC06, BCC07, and BCC08

-> Revised as suggested.

- Have authors tried a cocktail mix of the lytic phages to overcome the endogenous resistance of some of the species? That would be a beneficial screening that could easily translate into clinics due to the strain-specific nature of some phage activity. If it was not performed I recommend an assay with the 7 active phages together for screening, and if it was please describe it.

-> This is an excellent idea for a follow-on study, but we feel that it is beyond the scope of our current manuscript.